# Agronomic and Genetic Strategies to Enhance Selenium Accumulation in Crops and Their Influence on Quality

**DOI:** 10.3390/foods12244442

**Published:** 2023-12-11

**Authors:** Bingqi Zhou, Haorui Cao, Qingqing Wu, Kang Mao, Xuefeng Yang, Junxia Su, Hua Zhang

**Affiliations:** 1State Key Laboratory of Environmental Geochemistry, Institute of Geochemistry, Chinese Academy of Sciences, Guiyang 550081, China; zhoubingqi@mail.gyig.ac.cn (B.Z.); caohaorui@mail.gyig.ac.cn (H.C.); wuqingqing@mail.gyig.ac.cn (Q.W.); maokang@mail.gyig.ac.cn (K.M.); yangxuefeng@mail.gyig.ac.cn (X.Y.); sally15201@126.com (J.S.); 2University of Chinese Academy of Sciences, Beijing 100049, China

**Keywords:** Se, biofortification, crops, antioxidants, minerals, crop quality

## Abstract

Selenium (Se) is an essential trace element that plays a crucial role in maintaining the health of humans, animals, and certain plants. It is extensively present throughout the Earth’s crust and is absorbed by crops in the form of selenates and selenite, eventually entering the food chain. Se biofortification is an agricultural process that employs agronomic and genetic strategies. Its goal is to enhance the mechanisms of crop uptake and the accumulation of exogenous Se, resulting in the production of crops enriched with Se. This process ultimately contributes to promoting human health. Agronomic strategies in Se biofortification aim to enhance the availability of exogenous Se in crops. Concurrently, genetic strategies focus on improving a crop’s capacity to uptake, transport, and accumulate Se. Early research primarily concentrated on optimizing Se biofortification methods, improving Se fertilizer efficiency, and enhancing Se content in crops. In recent years, there has been a growing realization that Se can effectively enhance crop growth and increase crop yield, thereby contributing to alleviating food shortages. Additionally, Se has been found to promote the accumulation of macro-nutrients, antioxidants, and beneficial mineral elements in crops. The supplementation of Se biofortified foods is gradually emerging as an effective approach for promoting human dietary health and alleviating hidden hunger. Therefore, in this paper, we provide a comprehensive summary of the Se biofortification conducted over the past decade, mainly focusing on Se accumulation in crops and its impact on crop quality. We discuss various Se biofortification strategies, with an emphasis on the impact of Se fertilizer strategies on crop Se accumulation and their underlying mechanisms. Furthermore, we highlight Se’s role in enhancing crop quality and offer perspective on Se biofortification in crop improvement, guiding future mechanistic explorations and applications of Se biofortification.

## 1. Introduction

Se is an essential trace element for maintaining human health [1,2] that can form selenocysteine (SeCys) and serves as the active centre for various selenoproteins. These selenoproteins participate in several vital metabolic processes in the human body, such as antioxidant defence, thyroid hormone production, and cardiac metabolism [3,4]. Inadequate Se intake can impair various functions related to human health and lead to a weakened immune system, thyroid dysfunction, and decreased male fertility [5,6]. Optimal Se intake benefits human health by slowing the ageing process, reducing inflammation, and preventing oxidative stress [7,8]. Se intake levels in humans are contingent upon the concentration and form of Se in their diets [2]. It has been reported that the assimilation of organic Se from food is approximately 85–95%, while the assimilation efficiency of inorganic Se is approximately 40–50% [9]. Organic Se is absorbed more readily by the human body in a nonspecific manner [5,10]. It plays a more significant role in promoting human health. Therefore, consuming Se-rich foods, particularly those rich in organic Se, can effectively overcome Se deficiency.

Se-rich crops serve as the primary organic Se source for both humans and animals [11,12]. Their significance for human health cannot be overstated. The levels of Se absorption and accumulation in crops are closely linked to the genetic regulation of Se transport and metabolic mechanisms within crops [13]. Previous publications have discussed the process of Se uptake, transport, and metabolism of Se within crops [14,15]. In essence, crops initially absorb Se in the form of selenate, selenite, or organic Se compounds such as selenocysteine (SeCys) and selenomethionine (SeMet). Selenates are transported into crops by sulfate transporters, including SULTR1;2 and SULTR1, with SULTR1;2 being the main transporter involved in selenate uptake [16,17]. Early research indicated that crops absorb selenite through passive diffusion [18]. However, subsequent research has shown the participation of the silicon transporter OsNIP2;1 (Lsi1) aquaporin protein [19]. Furthermore, the phosphate transporter OsPT2 is also involved in the transport process of crops for selenite [20]. Upon entering the crop, selenate is initially reduced to selenite. This reduction process involves ATP sulfurylase (ATPS) and ATP reductase (APR) [21]. Subsequently, selenite is further reduced to selenides by selenite reductases or reduced glutathione, forming selenide [22]. Selenide is converted to SeCys through the action of cysteine synthase. However, improper incorporation of SeCys into proteins can be detrimental for crops. This is a primary reason why most crops do not accumulate significant amounts of Se. As a result, certain crops can synthesize the volatile compound dimethyl selenide (DMSe) from SeMet. They excrete it to reduce the biological toxicity caused by excessive Se accumulation. It is noteworthy that the aforementioned process represents the primary mechanisms of Se absorption and metabolism in plants. Most crops generally adhere to this process, although this is contingent on whether these crops possess the corresponding transporters or enzymes. Additionally, the outlined Se metabolic pathways are not exhaustive. Further research is still required to gain a comprehensive understanding.

Se biofortification aims to increase the bioavailable Se content in edible crops through diverse methods [23]. These methods encompass both agronomic and genetic strategies. Agronomic strategies for Se biofortification involve adjusting the concentration of Se fertilizers, changing the types of Se fertilizers, modifying the timing of Se application, and implementing agronomic management practices. For example, Finland initiated the addition of selenate to all agricultural fertilizers in the 1980s to enhance the Se content in crops [24]. Lidon et al. conducted the biofortification of rice using selenate and selenite as foliar fertilizers for four different genotypes [25]. The results indicated that, compared to selenate, selenite led to a greater accumulation of Se in grains. Hao et al. reported that applying Se during the heading stage in oats resulted in a higher Se content in oat grains compared to the stem-elongation stage [26]. However, the results of Se biofortification in Finland also show that crop uptake of Se from soil-applied Se fertilizers is typically less than 10% [11]. Therefore, to improve the efficiency of Se fertilization, agricultural practices commonly employ various agronomic management practices. These include crop rotation, intercropping, irrigation, and microbial inoculation.

Genetic strategies for Se biofortification aim to enhance the Se uptake and accumulation capabilities of target crops. This can be achieved through either conventional breeding or transgenic techniques. These approaches are considered a sustainable, long-term method [27]. Traditional breeding methods involve combining Se-rich varieties through genetic methods, such as hybridization, to enhance the Se-rich capacity of the target variety. In contrast, transgenic techniques have the potential to efficiently introduce genes from naturally Se-rich crops into staple crops. This approach can overcome the limitations associated with the single-species genetic pool. However, transgenic technology still raises significant ethical concerns in some countries, which makes its widespread adoption and application challenging [28].

At optimal concentration conditions, Se enhances crop growth and antioxidant activity, increases the levels of various macro-nutrients and beneficial mineral elements, and effectively reduces the content of heavy metal elements [29,30,31]. For instance, foliar Se application effectively increased the content of Ca, Mg, and Zn in purple-grained wheat seeds while simultaneously reducing the accumulation of Cr, Pb, and Cd [32]. In a recent study, the biofortification of dragon fruit (*Hylocereus undatus*) with Se demonstrated an improvement in antioxidant activity [33]. This enhancement led to the accumulation of bioactive compounds, including phenolic acids, flavonoids, and betalains in the pulp. This effectively strengthened dragon fruit’s resistance to pests and diseases, leading to improved crop yield and nutritional value. This may be attributed to Se’s activation of the crop’s oxidative stress defence mechanisms, such as the restoration of membrane enzymes [34]. The antagonistic effect of Se on various heavy metal elements in crops may be attributed to its capacity to decrease the bioavailability of heavy metal elements. Additionally, Se inhibits the crop’s ability to absorb and transport heavy metal elements. However, this promoting effect of Se biofortification is not universal and depends on the crop species, genotype, and the Se fertilization strategy used.

This review provides a comprehensive summary of the impact of various Se biofortification strategies on Se accumulation and quality in crops over the past few decades. In particular, we have meticulously documented the effects of different Se biofortification methods, including Se fertilization strategies, agronomic management approaches, breeding techniques, and transgenic technologies, on the Se content and beneficial nutrient levels in crops. Additionally, we explore potential mechanisms underlying these effects with the aim of offering guidance for subsequent research endeavors and agricultural practices. In conclusion, we propose evaluation criteria for future Se biofortification efforts. Initially, the principal objective of Se biofortification should center on regulating the Se content within the edible portions of crops, ensuring that it falls within a suitable concentration range. This adjustment is contingent upon the varying dietary needs for different crop types [35]. Secondly, the promotion of crop yield by Se should be a crucial criterion for measuring the success of Se biofortification, considering its implications for global food security. Lastly, given the increasing prominence of hidden hunger, the role of Se in promoting the accumulation of beneficial nutrients in crops should be emphasized. This can effectively enhance the competitiveness of Se-biofortified foods in the market.

## 2. Agronomic Strategies for Se Biofortification

Se biofortification agronomic strategies can be categorized into Se fertilizer strategies and agronomic management strategies (Figure 1). In this section, we initially summarize the crucial role of Se fertilizer strategies, including how to adjust the type [36,37,38] and dosage of Se fertilizers [39,40], and the application site [32,41] and timing of Se [42,43]. Subsequently, we consolidate and discuss the impacts of agronomic management strategies on Se biofortification, encompassing crop rotation [44,45,46] and intercropping [47,48,49] and soil [50,51,52] and water management [53,54,55], as well as microbial inoculation [56,57,58]. As the most accessible methods for farmers, agronomic strategies play an irreplaceable role in Se biofortification practices.

### 2.1. Se Fertilization Strategies

#### 2.1.1. Effects of Se Species and Dosages on Crop Se Accumulation

Selenate and selenite remain the most commonly used Se fertilizers, although some studies have also utilized organic Se and nano-Se. For most crops, selenate is more readily absorbed than selenite [60,61,62]. According to the multilevel prediction model by Ros et al., the addition of an appropriate dosage of selenate (14 g ha^−1^) or selenite to crop roots resulted in a significant increase in the total Se content in the crops, with changes of +590% and +260%, respectively (Figure 1) [59]. The higher absorption efficiency of selenate by crops might be due to its reduced tendency to adsorb to the soil and its easier transport from the roots to the stems and leaves [63,64]. Previous research has suggested that crops supplied with selenite primarily accumulate organic Se, while those supplied with selenate mainly accumulate selenate [65]. The conversion of selenate to selenite appears to be the step that limits the rate of selenate assimilation into organic Se [66]. D’Amato et al. treated rice seeds with 135 mg L^−1^ selenite and selenate [67]. The results revealed that in the shoots treated with selenite, the organic Se content was 23.8 μg g^−1^ DM, accounting for 93.8% of the total Se content. In comparison, in rice seedlings treated with selenate, the organic Se content was 24.9 μg g^−1^ DM, representing 32.5% of the total Se content. This suggests that although the conversion of selenate to organic Se in crops is relatively low, the higher absorption of selenate by crops results in a greater accumulation of both organic and inorganic Se. In general, inorganic Se is considered toxic to humans, while organic Se is beneficial [68]. In summary, selenate seems to function as a more effective Se fertilizer, but attention should be directed towards the possible health hazards associated with the excessive accumulation of inorganic Se in crops. Selenate is a more efficient Se species, but it raises concerns about potential health risks. The application of selenite may provide a safer approach for organic Se food production. However, due to selenite’s higher soil adsorption [64] and lower crop uptake and translocation rates, its biofortification efficiency might be insufficient.

In general, the level of Se accumulation in crops is positively correlated with the concentration of Se fertilizer [32,40,69]. However, determining the optimal Se fertilizer dosage solely based on the total Se content in crops is an erroneous approach. Factors such as Se fertilizer utilization efficiency and the form and concentration of accumulated Se in crops must also be taken into consideration. For instance, in Hu et al.’s study on the Se biofortification of *H. ericium erinaceus*, it was observed that under selenate treatment conditions ranging from 0.5 to 80.0 μg g^−1^, the bioconcentration factor of Se in mushroom fruiting bodies exceeded 1. However, under the treatment condition with 100 μg g^−1^ selenate, the bioconcentration factor dropped below 1 [61]. Another study demonstrated that when the Se fertilizer dosage increased from 50 g Se ha^−1^ to 100 g Se ha^−1^, the proportion of organic Se in peanuts decreased from 88.2% to 80.1% [39]. These studies indicate that while increasing Se fertilizer concentration can boost Se accumulation in crops, it may also lead to decreased Se fertilizer and organic Se conversion efficiency, potentially causing excessive Se retention and posing a threat to environmental health. Furthermore, considering the narrow concentration range of Se that is beneficial for human consumption, it is advisable to select appropriate Se fertilizer concentrations for different crops.

#### 2.1.2. Effect of Crop Fertilization Site and Time on Se Accumulation

Based on the mechanisms of Se uptake in crops, the root application of Se and the foliar application of Se are the two most common application methods for Se supplementation. Research has shown that the root application of Se can substantially boost Se concentration in crops [70]. Liu et al. revealed that within a Se concentration range that did not affect the biomass and yield of wheat, root application of Se could increase the Se content in wheat kernels by more than 17 times [40]. These studies demonstrate that the root application of Se is highly effective for improving crop Se accumulation, making it a practical and viable approach. However, the utilization rates for root-applied Se in crops is generally low, and long-term root application may lead to increased soil toxicity [71]. Comparatively, foliar Se treatment can utilize Se fertilizers more effectively [59]. Se can enter the crop through the leaf, reducing the environmental impact during this process and shortening the distance for Se transport to crop seeds [72]. Xia et al. reported that foliar-applied Se in purple-grain wheat led to higher Se accumulation in grains, stems, leaves, awns, and spike tips as compared to root application [32]. Currently, foliar Se application has become a commonly used approach for Se biofortification in staple crops such as wheat and rice [37,62,73,74,75,76].

The transport of Se within crops is closely related to the growth stage of the crop. It is difficult for rice to oxidize and transport absorbed selenite to the aboveground parts of the plant during the early growth stage, but it can effectively transport and accumulate Se in the grains during the tillering or heading stages [77]. Li et al. applied Se to Foxtail millet (*Setaria italica* L.) during both the grain-filling and jointing periods. The results indicated that millet treated with Se during the grain-filling period accumulated more Se in the grains [43]. Additionally, Hao et al. reported that applying Se during the heading stage of oats led to increased Se accumulation in the grain portion as compared to the flowering stage [26]. Wang et al. suggested that foliar Se application to wheat during the grain-filling stage, as opposed to the earlier growth stages or other stages, results in the highest grain Se content [62]. Additional investigations have confirmed a noteworthy reduction in Se content in wheat leaves subjected to Se treatment during the grain-filling period as compared to the vegetative growth stage [78]. Collectively, these studies suggest a preferential transfer of Se from leaves to grains during grain development stages, such as flowering and grain-filling. The application of Se during these stages has been associated with increased Se content in crop grains.

Overall, Se fertilization is a straightforward and effective method for Se biofortification. However, poorly planned Se application methods do not align with the principles of “precision agriculture” and can lead to some degree of Se resource waste and soil pollution issues. Therefore, we have summarized the impacts of different Se fertilization strategies on crop Se accumulation with the intention of providing guidance for future research and Se application schemes in agricultural production.

### 2.2. Agronomic Management Strategies

#### 2.2.1. Crop Rotation and Intercropping

Crop rotation systems can enhance agricultural ecosystem functions, contributing to improving soil fertility, maintaining soil structure, and increasing soil microbial diversity [79]. As a traditional agricultural practice, crop rotation has been demonstrated for decades to increase the bioavailability of micronutrients such as Fe, Cu, Zn, and Mn [80,81]. Recently, several studies have demonstrated the crucial role of crop rotation in Se biofortification [82]. For example, in a barley-red clover-potato rotation system, applying Se through foliar spray to the barley alone increased the Se levels in the barley grains and stems, red clover leaves, and potato tubers [83]. In a foliar application of Se within the crop rotation system involving winter wheat and summer maize, maize almost completely consumed the residual Se in the soil [44]. These findings demonstrate that variations in Se uptake and accumulation among crops in rotation systems, as opposed to monoculture systems, lead to the effective utilization of the residual Se in the soil. Additionally, the increase in crop species diversity and microbial diversity in rotation systems may contribute to the higher levels of bioavailable Se in the soil [71].

Although intercropping is an ancient agricultural method, it still contributes to addressing some major challenges of modern agriculture, including soil degradation, crop yield decrease, and environmental degradation [84]. According to the stress gradient hypothesis [85], as environmental stress increases, crops are more likely to exhibit net positive (facilitative) interactions with other crops [86,87]. Therefore, the introduction of Se as a nonessential element for crops might lead to facilitative interactions between crops in intercropping systems. Recently, Tang et al. studied the impact of intercropping on Se levels in bok choy, lettuce, and radishes. The findings revealed that, compared to monoculture techniques, intercropping elevated the Se contents in the edible parts of bok choy and radishes [47]. Lin et al. reported that intercropping cherry tomatoes with three wild eggplant species of the Solanum genus (diploid, tetraploid, and hexaploid) resulted in 13.73%, 17.49%, and 26.50% increases in Se content in cherry tomato seedlings compared to cherry tomato monoculture [49]. Similarly, Pan et al.’s study indicates that intercropping three varieties of eggplant (red, green, and black eggplant) can enhance the Se content in the seedlings of red and black eggplants [48]. Additionally, intercropping with two varieties also contributes to Se biofortification. For example, intercropping red eggplant with green eggplant can increase the Se content in green eggplant seedlings. Intercropping enhances the energy exchange between the crops, the soil, and microorganisms. This alteration in the bioavailability of Se in the rhizosphere influences the absorption of Se by crops [48,88]. Moreover, research suggests that intercropping can enhance crop resistance to Se. This, in turn, increases the uptake and accumulation of Se in crops [49].

#### 2.2.2. Soil and Water Management

Soil porosity plays a crucial role in influencing root growth and function, the transport of solution ions to the crop root surface, and processes like the leaching of solutes in the rhizosphere [89]. Soil compaction is a process that reduces soil porosity and increases bulk density, leading to changes in soil chemical properties and biodiversity [90]. Appropriate soil compaction can reduce soil porosity, increase contact between the rhizosphere surface and the soil solution, and enhance the crop’s absorption of trace elements from the soil [91,92]. For instance, Zhao et al. suggested that restricted root growth in compacted soil may contribute to a significant decline in Se concentration in wheat grains [55]. Excessive compaction, however, can result in oxygen deficiency in soil pores. This leads to the reduction of biologically available Se to forms that plants cannot absorb [93]. In agricultural practices, to prevent severe soil compaction, soil amendments [94] or tillage methods [95] can be employed to regulate soil porosity. Tillage significantly influences the physical and chemical properties of soil [96] as well as the form and distribution of elements [95,97]. Some studies have explored the role of tillage in Se biofortification. Lessa et al. discovered that the tillage process reduced Se adsorption in the soil [98]. It brought Se from deeper soil layers to the surface, thereby increasing the crop-available Se content. Similarly, Ozpinar et al. reported varying degrees of Se concentration increases in all tissues of corn under three tillage systems (plow, rotary, and chisel) [52]. However, conflicting conclusions arise from Lopez-Bellido et al.’s work, suggesting that no-till systems do not induce stronger soil-reducing conditions than traditional tillage systems [99]. Tillage systems seemingly do not affect wheat’s Se uptake and accumulation. These contradictory findings may be related to soil moisture content, organic matter content, and biodiversity. Previous research indicates that soil moisture contributes to maintaining soil structure and enhancing soil resistance to compaction [100]. The presence of water promotes substance cycling in the soil, potentially influencing Se ‘s chemical properties in the soil and the process of crop Se absorption. Additionally, soil organic matter and soil biology are crucial factors that influence soil redox conditions. These factors are likely essential reasons for the variations in the Se biofortification effects observed in different tillage studies. It can be concluded that soil compaction/tillage can influence Se biofortification in various ways (increasing/decreasing bioavailable Se content or promoting/inhibiting crop growth) [71]. Further work is needed in order to elucidate the geochemical behaviour of Se in these agronomic management practices and reveal the detailed processes by which agronomic management influences Se biofortification.

As water scarcity becomes an increasingly pressing issue, water management has emerged as a focal point in agricultural research. Irrigation is one of the most common and impactful water management measures in agricultural systems [101]. Irrigation provides a temporary Se reservoir for crops, significantly influencing their Se concentration and forms [102,103]. Zhao et al. found that irrigation during the wheat growth period doubled its grain yield [55], while the Se concentration decreased by 10 times, which may be due to the increased yield and leaching of soil-available Se caused by irrigation [104]. Recent research supports this notion. Ma et al. conducted an eight-year monitoring and investigation process of high-Se soils and adjacent soils, revealing that irrigation water led to Se leaching and migration in the soil [105]. However, in some studies, the content of bioavailable Se in the soil has increased due to changes in the soil redox conditions and pH caused by irrigation. For instance, Wang et al. found that aerobic irrigation increased soil pH, redox potential, and bioavailable Se content to a certain extent, resulting in an increase in the Se content in crop grains [106]. Overall, the impact of irrigation on Se biofortification is complex. Aerobic irrigation leads to the conversion of Se into more bioavailable forms, but Se dissolution and migration during irrigation reduce soil Se content. Therefore, further research is needed to elucidate the mechanisms of irrigation’s impact on soil Se contents under different soil conditions.

#### 2.2.3. Microbial-Assisted Biofortification

The low bioavailability of Se fertilizers in agronomic biofortification strategies has sparked significant interest in microbial-assisted Se biofortification. Rhizosphere bacteria and arbuscular mycorrhizal fungi (AMF) are the two most commonly employed microorganisms for Se biofortification, as they can effectively enhance crops’ Se uptake [107,108].

AMF are commonly associated with crop roots. They facilitate the Se supply to crops through an extensive mycelial network [109]. AMF also modify the chemical compounds and pH of crop roots in the rhizosphere via metabolic activities, such as the secretion of extracellular phosphatase, and ultimately enhance the Se bioavailability in root soil [110,111]. Chen et al. found that the inoculation of AMF notably enhanced the content of organic Se in rice [112]. Additionally, the enhanced Se uptake in crops through AMF is associated with their capacity to encode sulfate transport proteins. AMF induce the production of specific sulfate transport proteins in crops, leading to increased Se uptake [113].

Bacteria play a pivotal role in the Se biogeochemical cycle [114], especially within the rhizosphere of crops. Bacteria capable of accumulating Se from the soil and supplying it to crops are referred to as Se bacteria [115,116]. Se bacteria utilize metabolic processes to chemically transform Se (via oxidation, reduction, or methylation) and assimilate it within their cells through respiration [117]. Generally, based on the location of bacterial inoculation, Se bacteria can be classified into rhizosphere bacteria and endophytes. Yasin reported that the application of YAM2 (a rhizobacterial strain with 99% similarity to *Pichinotyi bacillus*) in wheat kernels and stems increased Se levels by 167% and 252%, respectively [118]. The study by Feng et al. demonstrated that a bacterial consortium synthesizing iron carriers led to a 68.7% increase in soil-bioavailable Se and a 92.2% improvement in crop Se-uptake efficiency [57]. These findings indicate that Se bacteria not only promote the production of bioavailable Se in the rhizosphere but also enhance crop Se-uptake capabilities simultaneously. In comparison, endophytic bacteria have a weaker regulatory effect on Se in crop rhizosphere soil, which might not be conducive to Se accumulation in crops. However, endophytic bacteria have stronger associations with crops, are protected by crop tissues, and have more stable growth environments. Consequently, they often provide greater benefits to crops in terms of promoting growth, nutrient accumulation, disease suppression, and environmental stress resistance [119,120]. For instance, Trivedi reported that soybean crops treated with endophytic Se bacteria MGT9 showed improved growth under drought conditions, and the Se biofortification effect in soybean crops increased by 7.4 times [115].

These studies indicate that the microbial inoculation of crops can enhance crop Se-transport capacity and the Se forms in the crop rhizosphere. This effectively compensates for the limitations of Se fertilization strategies and represents a promising auxiliary technology for Se biofortification.

## 3. Genetic Strategies for Se Biofortification

The utilization efficiency of Se fertilizers in agronomic practices is influenced by various factors, with genetic constraints in crops being a key factor. Researchers in Se genetic biofortification aim to cultivate specific crop species or genotypes capable of accumulating adequate Se in their edible parts by harnessing gene strategies. This involves a comprehensive understanding of the mechanisms of Se absorption and metabolism by crops [14]. Generally, gene-based Se biofortification strategies involve enhancing the genes of target crops through traditional breeding and genetic engineering techniques. This aims to achieve improved Se uptake and accumulation in the target crop [121]. Traditional breeding methods have attracted a lower research interest but have a higher success rate, making them reliable biofortification approaches [121]. However, breeding technology is characterized by a lengthy varietal development period and faces challenges in surpassing the limitations associated with genetic resources. Transgenic technology has the potential to overcome genetic limitations by introducing genes from other species. However, its high cost and ethical concerns make it challenging to widely implement at the moment.

### 3.1. Breeding Techniques for Gene Improvement in Crops

Traditional breeding methods involve genetic approaches, such as hybridization, to combine varieties rich in the target nutrient in order to improve the genetics of various crops [122] (Figure 2a,b). There is wide genetic variability in the Se concentrations in potato tubers [123], soybean seeds [124], rice [125,126], wheat grains, and various leafy vegetables [127]. For example, Zhao et al. [128] isolated a rice mutant, cadt1, from an ethyl methanesulfonate (EMS)-mutagenized population of indica rice varieties. The mutation in the OsCADT1 gene led to an increase of 2.4 times in the maximum rate of selenate absorption in cadt1, indicating a significant enhancement in selenate uptake in the mutant. Souza et al. conducted Se biofortification studies on 20 Brazilian wheat varieties selected from a wheat breeding program (*Planaltina*, GO, Brazil). The results revealed a genetic variation of a 1.5-fold difference in grain Se concentration among these wheat lines [126]. Thavarajah et al. found significant genetic variation in the Se concentrations (425–673 μg g^−1^) in 19 different varieties of lentils (*Lens culinaris* L.) [129]. Similarly, Rahman et al. observed significant genotype differences in seed Se concentrations and total Se yields among different genotypes of lentils planted at four locations, with no significant genotype × location interactions [130]. This suggests that enhancing Se biofortification through breeding to improve lentils traits is feasible.

Generally, traits related to the Se uptake in crops are quantitative traits, and these traits are under genetic control [131]. Quantitative trait loci (QTL) mapping can break down complex quantitative traits into several genetic loci, which helps to identify Se concentration QTL. Studying the genetic characteristics of different crops and identifying the QTL related to Se uptake and accumulation is crucial for crop breeding and Se biofortification. Se-concentration QTL have been identified in various common crops, including rice [132,133], wheat [134], and lentils [135] (Table 1). A recent study in a recombinant inbred line (RIL) population derived from a cross between Arabidopsis thaliana genotypes Ler-0 and Col-4 identified genetic loci on chromosomes 1, 3, and 5 that are associated with selenate tolerance [136]. This discovery aligns with the outcomes observed in the hybrid experiments conducted on *Arabidopsis thaliana*. In recent times, genome-wide association studies (GWAS) have been employed to pinpoint the Se concentration QTL in crops, notably in rice [137], wheat [138], and legumes [139]. GWAS is based on detecting the associations between genotypes and correlated traits based on linkage disequilibrium in a sample population [140]. For example, Zhang et al. detected three QTL for the Se concentration in rice grains from the entire genome of 698 accessions, which constituted two subsets (indica/Xian, X-set, and japonica/Geng, G-set) [141]. The application of QTL mapping and GWAS rapidly identifies genetic loci related to Se concentration traits, significantly enhancing the efficiency of traditional breeding and advancing its research and application in Se biofortification.

### 3.2. Transgenic Technology Expanding the Traditional Crop Gene Pool

Traditional breeding techniques depend on the genetic diversity of crop varieties, and incorporating desirable traits into target varieties takes many generations [146]. In cases of limited or absent genetic variations in Se uptake across diverse crop varieties, transgenic technology can be used to introduce genes from other species into crops to obtain varieties capable of Se enrichment [147].

Transgenic technology can boost the Se absorption in crops by overexpressing the genes associated with Se uptake and enhancing the crop’s resilience, storage, and volatility of Se. This process involves augmenting the conversion of selenate into SeCys and the alterations of SeCys into volatile dimethylselenide, SeCys methylation, and the conversion of SeCys into elemental Se (Figure 2) [148]. Li et al. simultaneously overexpressed the genes encoding the human selenocysteine lyase (HsSL) and selenocysteine methyltransferase from Astragalus bisulcatus in rice [149]. After the addition of selenate and selenite, the newly obtained transgenic rice exhibited an increase in fresh weight, with Se accumulation levels that were approximately 38.5% and 128.6% higher than those of the wild-type rice. Furthermore, it is noteworthy that Se hyperaccumulators, while not well-suited for direct biofortification, serve as ideal species for providing genes for Se biofortification. The SMT gene from the Se hyperaccumulator *Astragalus bisulcatus* serves as a typical example. Initially, researchers introduced the SMT gene into the non-hyperaccumulating plant Arabidopsis, confirming the feasibility of developing transgenic Se-enriched plants through SMT overexpression [150]. However, the low reduction rate of selenate by ATPS in Arabidopsis limited the synthesis of MeSeCys (Figure 2). Therefore, McKenzie et al. overexpressed the SMT in tobacco (*Nicotiana tabacum* L.) [151]. This resulted in a 2–4-fold increase in the total Se content, in which MeSeCys accounted for up to 20% of the total Se. Subsequently, Brummell et al. introduced the SMT gene SMT1 into tomatoes [152]. This enhanced the efficiency of the MeSeCys production. The newly obtained tomato genotype accumulated MeSeCys up to 16% of the total Se content. To identify more usable Se hyperaccumulator genes, Hung et al. treated seedlings of the Se hyperaccumulator A. *racemosus* with selenate and selenite and analyzed 125 selected Se-responsive candidate genes [153]. The results showed that selenate and selenite treatments induced expression levels of more than two-fold for nine and 14 genes, respectively (either induction or suppression). The expression levels of the newly induced gene CEJ367 were increased by 1920-fold and 579-fold, respectively. These identified and isolated genes can be utilized to create new transgenic crops. In comparison to traditional breeding techniques, transgenic technology is more sustainable and exhibits better enhancement effects, but has lower acceptance rates [121]. The primary factor limiting the application of transgenic technology is low public acceptance. Additionally, due to the need for identification and modification of target genes in transgenic technology and their expression in new species, the probability of successfully breeding new varieties with excellent traits is not high [154]. Finally, since different countries have varying regulatory standards for transgenic crops, going through these regulatory processes incurs high time and economic costs [155].

## 4. Impact of Se Biofortification on Crop Growth and Quality

Se biofortification is a promising agricultural practice. It not only increases the Se content in crops but also enhances their overall nutritional quality. This provides a range of essential nutrients and bioactive compounds that benefit human health [156]. The primary beneficial nutrients influenced by Se biofortification in crops include macronutrients (such as proteins, carbohydrates, lipids, and vitamins), antioxidant compounds (phytochemicals, carotenoids, flavonoids, vitamin C, etc.), and mineral elements (calcium, magnesium, potassium, manganese, zinc, iron, copper, etc.) [29,30,31]. Additionally, as a practical agricultural application technique, the practice of Se biofortification should take into account crop growth parameters and the presence of toxic heavy metal elements as crucial evaluation criteria.

### 4.1. Influence of Se Biofortification on Crop Growth

#### 4.1.1. Influence of Se Biofortification on Growth Status and Antioxidant Activity

Crop growth status is a crucial economic indicator in agricultural practices. The antioxidant activity of crops and growth parameters, such as root, stem, leaf length or biomass, and grain yield, typically reflect crop growth status. This section focuses on elucidating the impact of Se on crop antioxidant activity, which is a vital indicator of crop growth, particularly in the context of growth under stressful conditions.

The impact of Se on crop development is significantly linked to Se fertilization strategies and specific crop species or genotypes (Table 2). Field experiments with black-grained wheat have shown significantly higher yields in Se-rich regions than in Se-poor areas [40]. Saidi observed that when sunflower seedlings were exposed to 20 μM Cd treatment, the root and leaf fresh weights decreased by approximately 69% and 57.5%, respectively [157]. However, adding 5 μM Se increased the fresh mass of leaves and roots by approximately 48% and 59.50%, respectively. This could be due to Se activating the crop’s antioxidant defence mechanisms. Se-induced enhancement of antioxidant activity alleviates the inhibitory effects of heavy metals on crop growth. This includes the restoration of reduced photosynthesis caused by heavy metals, increased levels of various bioactive compounds in crops, and elevated antioxidant enzyme activity [34,158,159]. For example, research by Newman demonstrated that the oxygen radical absorbance capacity (ORAC) values of chives, basil, and cilantro increased by 152.2%, 68.6%, and 66.0%, respectively, under optimal sodium selenate conditions [160]. However, it is important to note that excessively high Se concentrations can harm crop growth, as many studies have shown [25,67,160,161]. For instance, D’Amato et al. found that treating rice seeds with 135 mg L^−1^ selenate resulted in a 39% reduction in shoot length and an 89% reduction in root length compared to the control group [67].The use of 405 mg L^−1^ selenate completely inhibited rice seed germination. In the study of potatoes by de Oliveira et al., a Se dose of 0.75 mg kg^−1^ increased the tuber yield by 4%, but when the Se dose increased to 5 mg kg^−1^, the tuber yield decreased by 17% [162]. Similar conclusions were drawn in research on wheat [40], rice [67], and mushrooms [163]. Mateus et al. reported that the foliar spraying of 20 mg L^−1^ sodium selenate on coffee leaves increased the chlorophyll content, while spraying 160 mg L^−1^ sodium selenate resulted in a chlorophyll content lower than that of the control group [164]. This suggests that a high concentration of sodium selenate may damage the photosynthesis capabilities of crops. High Se concentration induces oxidative stress, manifested by an increase in reactive oxygen species (ROS, such as H_2_O_2_) concentration in crops, which is a major reason for inhibited crop growth. When the reducing substances in crops (such as glutathione, thiols, iron oxyhemoglobin, and reduced coenzyme II) are insufficient to meet the simultaneous assimilation of Se and the quenching of ROS [34], excessive ROS can degrade enzymes and proteins in crops, leading to a decrease in crop antioxidant capacities and impaired growth [165,166,167]. For instance, it has been reported that the H_2_O_2_ content in coffee leaves treated with Se fertilization at doses of 10–40 mg L^−1^ was inversely correlated with the Se fertilizer concentration. However, as Se concentrations increased further (80–160 mg L^−1^), the H_2_O_2_ content gradually rose [164].

Apart from the Se dosage, the influence of Se biofortification on crop development, including antioxidant activity, is closely related to the type of Se fertilizer used. In most cases, the trends of the impacts of various Se forms on antioxidant activity (either positive or negative effects) are consistent, but the degree of impact varies (Table 2). It has been suggested that selenite may have a more significant toxic effect on crops than selenate because selenite is more readily assimilated into organic Se compounds such as SeCys and SeMet, potentially disrupting the typical production of proteins [168]. For instance, Groth reported that, compared to selenate, the application of selenite resulted in lower antioxidant activity in the ‘Golden Delicious’ apple genotype [30]. However, there are also studies that report different conclusions. D’Amato et al., for example, reported that under high Se concentrations, selenate application exhibited a stronger suppression of rice seedling growth, leading to delayed germination and development of rice seedlings [40].

In summary, Se biofortification generally promotes crop growth, but this stimulatory effect has a threshold, and excessive Se accumulation can impair crop growth. Ideally, Se biofortification applications should ensure that crops accumulate Se levels beneficial to human health while maximizing crop growth.

**Table 2 foods-12-04442-t002:** The effects of Se application on growth and macronutrient contents in different crops.

Species	Applications	Growth Parameters	Saccharides	Proteins	Fats	Antioxidant Activity	Antioxidant Enzyme	References
*Triticum aestivum*, L. cv. Baegjoongmil microgreens	Sodium selenite 0.125–1.0 mg L^−1^—hydroponic	Yield ↓* Microgreen weight ↓ Microgreen height ↓	–	–	–	NSA ↑0.25 mg L^−1^ ABTS NS; DPPH NS	SOD ↑0.125, 1.00 mg L^−1^	[169]
*Oryza sativa* L.’Ariete’ grains	Sodium selenate 30–300 g ha^−1^—foliar	–	Total Sugars ↑60–300 g ha^−1^	↑120–300 g ha^−1^	↑180 g ha^−1^	–	–	[25]
*Ipomoea aquatica* Forsk. ‘XGDB’	Selenite 0.2 mg L^−1^—Foliar	Biomass NS	–	NS	–	MDA NS	SOD NS; POD NS; CAT NS;	[29]
*Triticum aestivum* L. Shoots ‘BRS 264′	Sodium selenate 12–120 g ha^−1^ Se—foliar	Yield ↑	Total Soluble Sugars ↑ Sucrose ↑21, 120 g ha^−1^	NS	–	MDA NS H_2_O_2_ NS	APX ↑12–38 g ha^−1^; CAT NS; SOD NS	[78]
*Solanum tuberosum* L. Tubers	Sodium selenite 0.75–5.0 mg kg^−1^—soil	Production ↑0.75 mg kg^−1^ ↓3–5 mg kg^−1^	–	–	–	MDA ↑3.0–5.0 mg kg^−1^ H_2_O_2_ ↓	CAT ↑1.5–3.0 mg kg^−1^ SOD ↑1.5–5.0 mg kg^−1^	[162]
*Coffea arabica* red Itucaí Leaves	Sodium selenate 10–160 mg L^−1^—foliar	Yield ↑10–40, 120 mg L^−1^	–	–	–	MDA ↓; H_2_O_2_ ↓20–80 mg L^−1^, ↑160 mg L^−1^	CAT ↑20, 120–160 mg L^−1^ APX ↑20–160 mg L^−1^ SOD ↑20–160 mg L^−1^	[164]

Effects are in comparison to control groups without Se application. Note: * (↑) = increase; (↓) = decrease; NS = not significant.

#### 4.1.2. Influence of Se Biofortification on Small-Molecule Antioxidant Contents in Crops

In response to abiotic stressors such as Se, crops increase the synthesis of certain small-molecule antioxidants in order to cope with environmental changes [170]. These small-molecule antioxidants play a crucial role in crop growth processes and the regulation of ROS generation. This in turn effectively protects cells and tissues from chronic oxidative damage [171]. Huang et al. reported that germinating black soybeans (GBS) exposed to a Se concentration of 25 mg L^−1^ exhibited the highest total phenolic content, with GBS treated at 50 mg L^−1^ showing a 50.3% increase in total phenolic acid levels [172]. A significant correlation between Se fertilizer application and crop vitamin C content has been observed in various crops, including wheat [169], water spinach [29], Indian mustard (*Brassica juncea* L.) [173], and spinach [174], among others (Table 3). Crop pigments have gained increasing attention due to their environmentally friendly and non-toxic characteristics, along with other beneficial functions such as antimicrobial and antioxidant properties and enzyme induction [175]. Numerous reports have suggested that the addition of an appropriate Se supplementation can enhance the chlorophyll accumulation in crops and help mitigate chloroplast damage caused by environmental stressors [164,169,176].

Se, at its optimal concentration, facilitates the synthesis of small-molecule antioxidants in crops. This demonstrates its beneficial role in promoting human antioxidative defenses. However, Se’s promotion of these antioxidants exhibits a threshold effect. Therefore, Se fertilization strategies need to be adjusted to maximize the utilization of Se and antioxidants within crops.

### 4.2. Influence of Se Biofortification in Crops on Element Accumulation

#### 4.2.1. Influence of Se Biofortification on Human Health-Related Mineral Elements

Se biofortification is a feasible method for increasing the Se content in food, as well as for promoting other essential dietary elements such as calcium, potassium, sodium, magnesium, iron, copper, manganese, zinc, and more. The increased content of these beneficial dietary elements in Se-biofortified foods can render them more advantageous than Se supplements. Pannico et al. reported on the variation in mineral elements among four different microgreen genotypes under the same Se treatment conditions [177]. The results indicated that the levels of elements such as Fe, Zn, and Mn increased in cilantro and purple basil, while conversely, green basil and lettuce showed a decrease in the accumulation of Fe and Zn. Some studies have shown the influence of Se fertilization strategies on the accumulation of mineral elements in crops. For instance, Golubkina found that applying selenates to male spinach crops resulted in decreased Mg levels, with no significant difference in the Ca content, while the opposite was observed when selenite was used [174]. Sodium selenite application to wheat leaves and soil exhibited contrasting effects. Foliar Se application increased the Ca and Zn levels in wheat grains without significantly altering the Fe content. Conversely, soil Se application showed an opposite impact, resulting in no significant changes in the Ca and Zn content but exhibited a notable reduction in the Fe content [32]. Given these findings, some studies have also started to explore the relationships between these factors and the content of various mineral elements. A recent study indicated that the foliar spraying of selenates not only resulted in changes in the mineral elements in kale crops but also altered the quantitative relationships between the elements. For instance, the correlation coefficient between Se and Si significantly increased, while the coefficients between Zn, P, and Co notably decreased [178].

Se biofortification can yield either positive or negative impacts on the human intake of beneficial mineral elements. Further research is needed to identify which specific mineral elements are adversely affected and to what extent. Additionally, to mitigate potential reductions in human intake of mineral elements, Se biofortification should prioritize nonstaple foods.

#### 4.2.2. Influence of Se Biofortification on Toxic Heavy Metals

Heavy metals can cause damage to human or crop cells at extremely low concentrations. The incorporation of heavy metals into proteins may lead to the inhibition of protein activity or cause structural damage [179,180]. It can also induce cellular oxidation or reduce antioxidants, resulting in damage to crop cells. Several studies have demonstrated that Se, under the appropriate concentration conditions, can effectively reduce the accumulation of toxic heavy metals such as Cd, Hg, As, Pb, and Cr in crops and alleviate toxicity (Table 4). There have been a significant number of detailed studies [34,181,182,183,184] summarizing the mechanisms of Se protection in crops, which are briefly discussed in this paper. First, Se can alter the chemical forms of heavy metals in the soil, thereby reducing their bioavailability and decreasing the crop uptake [185]. Zhang found that, due to the stronger affinity of Se for binding with Hg compared to S, Se can replace the S atom in methylmercury-cysteine (MeHg–Cys) to form biologically unavailable methylmercury–selenocysteine (MeHg–Sec) complexes, significantly reducing the biological toxicity of Hg [186]. Second, Se can reduce the toxicity of heavy metals to crops by inhibiting the translocation of heavy metals from the roots to the aboveground parts and reducing their accumulation in the edible parts of crops [187]. Zhang et al. showed that elevated Se levels in rice soil led to a consistent reduction in the transfer of inorganic mercury and methylmercury to the aboveground parts of rice [188]. Additionally, the inorganic Se content in grains exhibited an inverse correlation with inorganic mercury and methylmercury. Further investigation revealed that Se supplementation resulted in a reduction of 26.52%, 37.90%, and 47.57% in the cadmium content found in the shoots, roots, and leaves of tomato seedlings, respectively [189]. This suggests that Se concurrently inhibits the processes of cadmium entering crops and translocating within crops. Additionally, the change in the content of heavy metals in Se-supplemented crops may primarily depend on the type of crop and the type of heavy metals rather than the type and dosage of the Se fertilizer (Table 4). For instance, soil-applied Se led to a decrease in Pb in purple-grain wheat (‘202w17′) [32] but had no effect on the grain Pb in three other wheat varieties (Xihei 88, Heidali, and Pubing 151) [40]. In experiments on the Se biofortification of lion’s mane mushrooms, it was found that the application of 40 μg g^−1^ of selenite and SeMet both resulted in a decrease in As content in the fruiting bodies but increased the Cr content [61].

Assessments concerning the impact of Se biofortification on the accumulation of heavy metals in edible crops have reported a significant decrease in Cd, Hg, As, Pb, and Cr contents (Table 4). This suggests that Se biofortification restricts the entry of heavy metals into the food chain through crops in most cases, effectively reducing heavy metal stress in the soil-crop-human system.

## 5. Conclusions and Perspective

In recent years, researchers have made significant progress in optimizing Se biofortification methods and expanding the species that can be fortified. They have also preliminarily elucidated the response mechanism of crops to exogenous Se. In addition, more and more studies have shown that the significance of Se biofortification is not limited to supplementing human Se. The impact of Se biofortification on other nutrients in crops indicates that Se biofortification has potential health value for humans. Therefore, the choice of Se biofortification strategy should be determined according to actual needs. For most Se-deficient populations living in underdeveloped and low-Se areas, increasing the organic Se content of crops is still the primary goal. Therefore, it is appropriate to choose staple crops for Se biofortification and use Se fertilizers to quickly increase soil and crop Se content. For high-income populations in developed areas, the impact of Se biofortification on other nutrients in crops may be more worthy of attention. Due to the large differences in the impact of Se on different nutrients in crops, non-staple crops should be selected for Se biofortification to avoid nutritional imbalances. The significance of this review lies in its potential to provide the feasibility and advantages of Se biofortification for different crop needs, which are conducive to expanding the application scenarios of Se biofortification. Future Se biofortification research should pay attention to reducing the excessive use of Se fertilizers and combining biotechnology to improve the absorption and accumulation capacity of crops for Se. In addition, the mechanism of Se-induced crop growth and quality changes are not yet clear, and it is necessary to combine botany, genetic engineering, and omics research to further reveal the non-biological stress tolerance mediated by Se. Finally, it is necessary to integrate clinical medicine for a comprehensive assessment of the health effects of Se biofortified crops on the human body.

## Figures and Tables

**Figure 1 foods-12-04442-f001:**
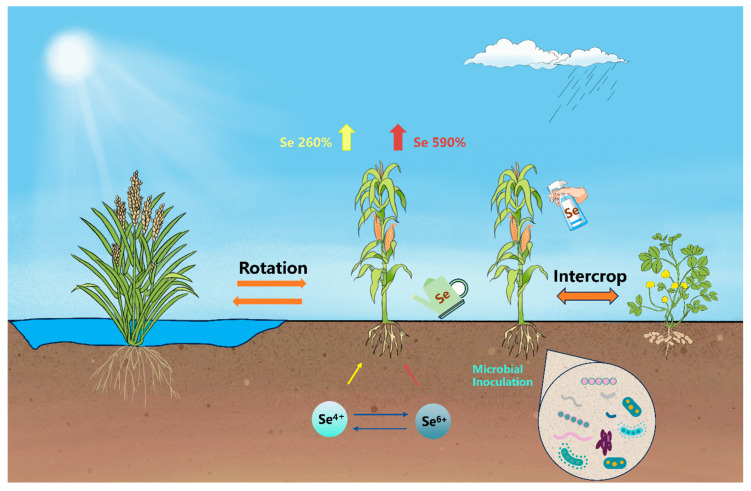
Agronomic strategies for Se biofortification. Note: The three crops from left to right are rice, corn, and peanuts; the data on crop absorption efficiency for selenate and selenite are derived from the meta-analysis by Ros et al. [59].

**Figure 2 foods-12-04442-f002:**
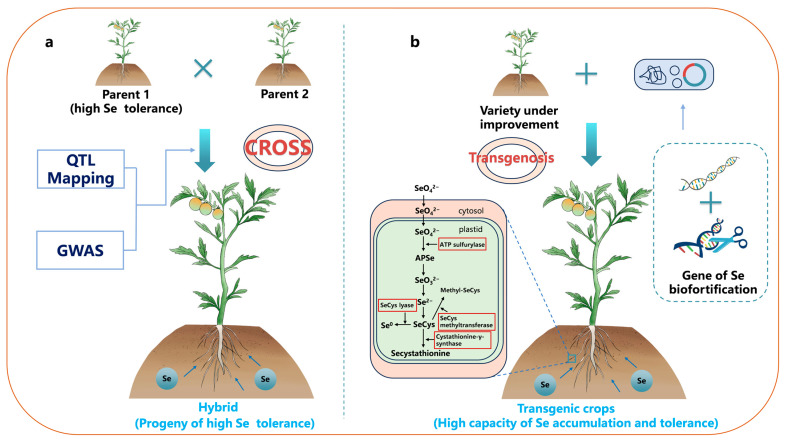
Genetic strategies for Se biofortification. (**a**) Breeding techniques and (**b**) Transgenic technology. Enzymes outlined with a red border are those overexpressed using transgenic technology. Note: QTL Mapping: quantitative trait loci mapping; GWAS: genome-wide association studies.

**Table 1 foods-12-04442-t001:** Mapping populations and QTLs for grain Se contents in different crops and related varieties.

Crop Types	Mapping Population		No. of QTL	References
	Cross	Type (Number)	Se	
*Lens culinaris* Medik.	PI 320937 × Eston	RIL (96)	36	[135]
*T. aestivum*, L.	TN18 × LM6	RIL (184)	16	[131]
*T. aestivum*, L.	*Triticum dicoccoides* × Langdon	F6 RIL (152)	15	[142]
*T. aestivum*, L.	SHW-L1 × Chuanmai 32	RIL (-)	4	[143]
*Oryza sativa* L.	*Oryza sativa* ssp. *indica* inbred variety 93–11 × *Oryza sativa* ssp. *indica* photo-thermo-sensitive male sterile line PA64s	RIL (132)	2	[133]
*Oryza sativa* L.	Bala (an *indica*) × Azucena (a *japonica*)	F6 RIL (105)	6	[144]
*Arabidopsis thaliana*	Ler-0 × Col-4	F8 RIL (96)	3	[145]

Se = Selenium content; RIL = recombinant inbred lines.

**Table 3 foods-12-04442-t003:** The impact of Se application on bioactive compounds and crop pigments in various crops.

Crop	Applications	Bioactive Compounds	Crop Pigments	References
*Oryza sativa* L.	Sodium selenite	Soluble Free Phenolic Acids (PA) ↑* 45–405 mg L^−1^	Total chlorophyll content (TChlC)↑15–45 mg L^−1^, ↓135–405 mg L^−1^	[67]
(Shoots of 10-Day Old Rice Sprouts)	15–405 mg L^−1^—plastic trays	Bound PA ↑45 mg L^−1^, ↓15, 135–405 mg L^−1^ Soluble Conjugated PA ↑	Total Carotenoid content (TCC) ↓45–405 mg L^−1^	
*Oryza sativa* L.	Sodium selenate	Soluble Free PA ↑	TChlC ↓45–135 mg L^−1^	
(Shoots of 10-Day Old Rice Sprouts)	15–135 mg L^−1^—plastic trays	Bound PA ↓	TCC ↓45–135 mg L^−1^	
*Fragaria × ananassa* cv. Fruits	Sodium selenate	Total phenolic content (TPC) NS; Total Flavonoids ↓	–	[69]
	10, 100 μM—hydroponic	Total Flavonols ↓		
*Brassica juncea* L. Leaves	Sodium selenate	Vitamin C ↑; Carotene NS	Chlorophyll a NS; Chlorophyll b ↑	[173]
	50 mg L^−1^	Flavonoids ↑		
*Coriandrum sativum* L.	Sodium selenate	Total polyphenols ↑16 μM	β-carotene ↑8 μM, ↓16 μM	[177]
	8, 16 μM—capillary mat	Lutein ↓		
*Ocimum basilicum* L. ‘green basil’	Sodium selenate	Total polyphenols ↑8 μM	β-carotene ↓	
	8, 16 μM—capillary mat	Lutein ↑8 μM, ↓16 μM		
*Spinacia oleracea* L.	Sodium selenate	Vitamin C NS	Chlorophyll a ↓; Chlorophyll b ↓	[174]
male crop Leaves	0.28 mM—foliar		TChlC ↓; Carotenes ↓	
*Spinacia oleracea* L.	Sodium selenite	Vitamin C ↑	Chlorophyll a ↑; Chlorophyll b ↑	
male crop Leaves	0.28 mM—foliar		TChlC ↑; Carotenes ↑	
*Spinacia oleracea* L.	Sodium selenate	Vitamin C ↑	Chlorophyll a NS; Chlorophyll b ↑	
female crop Leaves	0.28 mM—foliar		TChlC NS; Carotenes ↑	
*Spinacia oleracea* L.	Sodium selenite	Vitamin C ↑	Chlorophyll a NS; Chlorophyll b ↑	
female crop Leaves	0.28 mM—foliar		TChlC ↑; Carotenes ↑	

Effects are in comparison to control groups without Se application. Note: * (↑) = increase; (↓) = decrease; NS = not significant.

**Table 4 foods-12-04442-t004:** The effects of Se biofortification on the content of Cd, Pb, Cr, Hg, and As in different crops.

Crop	Application	Cd	Pb	Cr	Hg	As	References
*Triticum aestivum* L.	Se ore powder	– *	NS	NS	NS	NS	[40]
‘Xihei 88′ Grain (black-grained wheat)	1080–4320 g ha^−1^—soil						
*Triticum aestivum* L.	Se ore powder	–	NS	NS	NS	NS	
Heidali’ Grain (black-grained wheat)	1080–4320 g ha^−1^—soil						
*Triticum aestivum* L.	Se ore powder	–	NS	NS	NS	NS	
*H. ericium erinaceus* fruiting bodies	Selenate	NS 40 μg g^−1^	–	NS 40 μg g^−1^	↑40 μg g^−1^	NS 40 μg g^−1^	[61]
	0.5–200 μg g^−1^—substrate						
*H. ericium erinaceus* fruiting bodies	Selenite	↓40 μg g^−1^	–	↑40 μg g^−1^	NS 40 μg g^−1^	↓40 μg g^−1^	
	0.5–200 μg g^−1^—substrate						
*H. ericium erinaceus* fruiting bodies	SeMet	NS 40 μg g^−1^	–	↑40 μg g^−1^	↓40 μg g^−1^	↓40 μg g^−1^	
	0.5–200 μg g^−1^—substrate						
*Brassica juncea* L. Leaves	Sodium selenate	↓	NS	NS	–	NS	[173]
	50 mg L^−1^						
*Triticum aestivum* L.	Sodium selenite	↓	↓	↓	–	–	[32]
202w17 (purple-grain)’ Grains	10 mg ml^−1^—foliar						
*Triticum aestivum* L.	Sodium selenite	↓	↓	NS	–	–	
‘202w17 (purple-grain)’ Grains	50 mg kg^−1^—soil						

Effects are in comparison to control groups without Se application. Note: * (↑) = increase; (↓) = decrease; NS = not significant.

## Data Availability

Data are contained within the article.

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
