# Peer review of "Agronomic and Genetic Strategies to Enhance Selenium Accumulation in Crops and Their Influence on Quality"

_foods, 2023, doi:10.3390/foods12244442_

Round 1
Reviewer 1 Report
Comments and Suggestions for Authors
The ms foods-2745224 entitled Progress in selenium biofortification and its effects on crop quality investigates an important topic but the authors have to revise it to make their ms fits into such high quality journal.
L10 first write Selenium and then Se, so the readers know it later in your ms. Selenium (Se) is an ….
L17 and L20 use abbreviations here for selenium
L20 use crop quality instead of plant quality
L31-33 Please use this citations “Influence of Nano Silicon and Nano Selenium on Root Characters, Growth, Ion Selectivity, Yield, and Yield Components of Rice (Oryza sativa L.) under Salinity Conditions”
L35 please use only the abbreviation of selenium, why authors sometimes use Se and some others use Se, check this issue in whole ms.
L45-48 Here I suggest the authors to use following citation “Integrated Application of Selenium and Silicon Enhances Growth and Anatomical Structure, Antioxidant Defense System and Yield of Wheat Grown in Salt-Stressed Soil”
L44-88 Why the authors focus on plant breeding? They should focus on the agronomic part in terms of Se biofortification. So why the following text? Traditional breeding methods combine Se-rich varieties using genetic methods such as hybridization to enhance the Se-rich capacity of the target variety. In contrast, genetic engineering technology has the potential to efficiently introduce genes from naturally Se-rich crops into staple crops, breaking through the limitations of the single-species genetic pool.
L116-127 use suitable citations for this text.
L117-119 where is this previous research? Previous research has shown that adjustments in Se fertilizer strategies can significantly impact the results of Se biofortification, including the type and dosage of Se fertilizers, as well as the location and timing of Se application
L208-220 Authors should use this ref “Optimizing Inputs Management for Sustainable Agricultural Development”
I want the authors to also add the mechanisms of Se and how it can affect water use efficiency and fertilizer use efficiency?
L223-224 In that text, authors can use this citation “Effect of Different Sowing Methods on Growth, Yield and its Components of Wheat Under Intercropping Patterns with Egyptian Clover Var. Fahl”
How you can explain that the application of Se can reduce the contact of other trace elements such as Cd, Pb, As etc.?
L237-238 I suggest always the authors can give some text for background and end the text with some previous studies, please use this “Additions of optimum water, spent mushroom compost and wood biochar to improve the growth performance of Althaea rosea in drought-prone coal-mined spoils”
L261-262 The authors here should add suitable citation, here is one “Interaction effects of nitrogen source and irrigation regime on tuber quality, yield, and water use efficiency of Solanum tuberosum L.”
Conclusions and perspective: In this section, you should focus on the most important thing that you want to show for the readers, so please kindly make it deeper and revise it.
Good luck
Comments on the Quality of English Languageminor edits are needed for English
Reviewer 2 Report
Comments and Suggestions for Authors
The paper entitled “Progress in selenium biofortification and its effects on crop quality” is a very valuable study on the biofortification of agricultural crops with selenium in order to improve their quality.
The authors have comprehensively described the importance of selenium in crop agronomic and genetic aspects and have identified the need for further research taking into account the economic efficiency of selenium biofortification and the cost reduction associated with this treatment.
There are some small areas that need to be modified in this article, please refer to the attachment.

Reviewer 3 Report
Comments and Suggestions for Authors
The manuscript offers valuable insights into selenium biofortification and its influence on crop quality. However, its organization and writing need to be improved. English editing is needed for the manuscript, and consideration should be given to breaking down long sentences for improved readability. The arrangement of the subtitles appears inappropriate; a reevaluation of the sequence is recommended.
Title
The title could be enhanced to "Agronomic and Genetic Strategies to Enhance Selenium Accumulation in Crops and their Influence on Quality”
Abstract
1- The abstract mentions the impact of Se on crop quality and its role in crop improvement, but it could be strengthened by briefly explaining why enhanced crop quality is important and how it relates to human health or agricultural practices.
2- Some sentences need to be condensed to improve conciseness and make the abstract more impactful. Consider revising or removing repetitive or redundant phrases to ensure every word contributes to the overall clarity and effectiveness of the abstract. For example Line 12 and line 15: long sentences please make it short and condensed.
3- Line 10 : Please add the full name of se selenium
4- Line 26 “providing guidance for future mechanistic” Replace it with guiding future mechanistic explorations.
5- Add crop quality to the keywords.
Introduction
The introduction is not organized and is not well presented, it requires to be improved
1- Line 50: explaining the absorption and accumulation of selenium in crops in good way. However, it would be helpful to clarify whether these processes are specific to certain crop species or if they apply universally.
2- Line 70: showed selenium biofortification strategies. it would be beneficial to provide specific examples or case studies to illustrate the efficacy of these strategies and their practical applications.
3- Line 90-100: It would be helpful to expand the effects of selenium biofortification on crop growth, antioxidant activity, and the content of macronutrients and heavy metal elements and provide more specific findings from relevant studies to support these claims.
4- Line 101-113: this paragraph mentioning the need for consistent evaluation parameters for assessing the impact of selenium biofortification. However, it would be valuable to suggest potential parameters or criteria that could be used to evaluate the success of biofortification efforts in future research.
5- There are a few grammatical errors and awkward sentence structures throughout the introduction that need to be addressed. Proofreading for clarity and coherence is recommended.
6- It would be helpful to provide a brief outline of the subsequent sections of the paper to give the reader a clear understanding of what to expect in the rest of the review.
Agronomic strategies for Se biofortification
1. Please add new subtitle about Se biofortification strategies in Plants
2. Line 117 add references
3. Line 144: D'Amato et al. To align with the journal's system, it is recommended to include the reference number immediately after citing the reference. For instance (Author et al., Reference #1). Kindly review the entire paper for verification.
4. Line 159: ???? Not sure about this statement
5. Line 175: it should be application sites not sites
6. Line 194: Please provide additional examples or studies supporting the optimal timing for Se application in different crops.
7. Line 207: Please provide more examples or studies demonstrating the impact of intercropping on Se biofortification in different crops. Moreover, it would be valuable to elaborate on the underlying mechanisms or factors contributing to the difference in Se uptake and accumulation among different crops.
8. Line 236: in the subtitle “Soil and water management”, the conflicting conclusions regarding the impact of soil compaction and tillage on Se biofortification are mentioned. However, it would be valuable to discuss the potential reasons for these discrepancies, such as variations in experimental conditions, crop species, or soil types. This would enhance the understanding of the topic.
9. Line 323: please provide more specific examples of crops and their respective genetic improvements for Se accumulation to illustrate the effectiveness of traditional breeding methods.
10. Line 341: It’s better to make new table include List of QTLs identified for Se biofortification in different crops including population type and size, number of total QTLs and explained variations.
11. Line 385: please explain the potential drawbacks or challenges associated with this approach. Addressing concerns such as regulatory considerations, potential ecological impacts, or unintended effects on crop traits would provide a more balanced perspective on the use of transgenic technology for Se biofortification.
12. Line 378: can you provide more examples of Se hyperaccumulators and the specific genes they contribute for Se enrichment in transgenic crops? This would highlight the significance of biodiversity in developing effective transgenic approaches for Se biofortification.
13. Line 387 the subtitle “Impact of Se biofortification on crop quality” could be change to Impact of Se biofortification on crop growth and quality
14. The sequence of the subtitles is not appropriate please review it.
15. Line 428: Does the gradual increase in H2O2 content at higher Se concentrations have any specific consequences for crop growth or physiological processes?
16. Line 436: please provide more information on the specific adverse effects observed in crops at high Se doses, such as changes in plant morphology, reduced photosynthetic efficiency, or altered nutrient uptake. Additionally, discussing the potential mechanisms underlying the pro-oxidative effect of high Se concentrations and its impact on crop growth would provide a more comprehensive understanding of these effects.
17. Line 454: rephrase to be “These small-molecule antioxidants play a crucial role in crop growth processes and the regulation ROS generation.
Conclusions and perspective
To strengthen the conclusion, consider providing specific recommendations for fortification methods and plant species selection. Additionally, discussing potential collaborations and interdisciplinary approaches can enhance the practical implications of the conclusions.
Comments on the Quality of English LanguageEnglish editing is needed for the manuscript, and consideration should be given to breaking down long sentences for improved readability.
Round 2
Reviewer 3 Report
Comments and Suggestions for Authors
The authors did a good job of revising the review article .
Comments on the Quality of English LanguageMinor editing of English language required